# Factors Associated with Attitudes towards Rejecting Intimate Partner Violence among Young Adults in Malaysia

**DOI:** 10.3390/ijerph19095718

**Published:** 2022-05-08

**Authors:** Wan Soliha Wan Mohd Hanafi, Tengku Alina Tengku Ismail, Anis Kausar Ghazali, Zaharah Sulaiman, Aziah Daud

**Affiliations:** 1Department of Community Medicine, Universiti Sains Malaysia, Kota Bharu 16150, Kelantan, Malaysia; wansoliha86@student.usm.my (W.S.W.M.H.); aziahkb@usm.my (A.D.); 2Biostatistics and Research Methodology Unit, Universiti Sains Malaysia, Kota Bharu 16150, Kelantan, Malaysia; anisyo@usm.my; 3Women’s Health Development Unit, School of Medical Science, Universiti Sains Malaysia, Kota Bharu 16150, Kelantan, Malaysia; zaharah@usm.my

**Keywords:** intimate partner violence, young adults, attitudes, acceptance, disclosure

## Abstract

Intimate partner violence (IPV) is a serious public health issue, which is health threatening across all age groups, including young adults, and makes them and vulnerable. The rejection of IPV and willingness to disclose IPV as part of the solution are important as they correlate to this concealed violent behavior. This study aimed to investigate determinants of attitudes towards rejecting IPV among young adults. A cross-sectional study was carried out on 405 young adults attending premarital courses who were selected using purposive sampling. A validated questionnaire (MYPAIPVQ) was used as the study instrument. Logistic regression analyses were performed to test for associations between sociodemographic characteristics and relationship status with attitudes towards IPV. Attitudes towards rejecting IPV included not accepting IPV and have the willing to disclose it. About half of the premarital young adults (50.4%) had attitudes towards rejecting IPV. In the regression analysis, age (AdjOR 1.12), female (AdjOR 2.49), self-employed (AdjOR 0.20), and drama as sources of information (AdjOR 3.66) were significantly associated with attitudes towards rejecting IPV. The findings have potentially important implications for interventions aimed at preventing violence among the young adult population as they are vulnerable to being involved in IPV in the future.

## 1. Introduction

Intimate partner violence (IPV) is defined as any violent and abusive behavior within an intimate relationship that causes physical or psychological harm to those in the relationship, perpetrated by the intimate partner. Although the above issue has been attracting much attention worldwide, acceptance of IPV is worrying as it leaves devastating consequences. It is reflected by a high prevalence of violence globally, up to 30.0% of the population, significantly in WHO African and Eastern Mediterranean countries [1]. Prevalence varies throughout Southeast Asian countries, with 34.3% in Timor Leste, 14.8% in the Philippines, and 13.7% in Cambodia. It was reported to be around 24% in high-income nations such as Europe and the Western Pacific [1]. In Malaysia, the prevalence estimates for lifetime IPV range widely from 8% in a national household survey to 87% in women’s shelters [2,3].

In 2021, statistical records from the Royal Malaysia Police Department showed that reported violence cases increased from 2500 cases in 2003 to 5260 cases in 2020 [4,5]. The increasing number of these reported cases require serious attention. However, it is only the tip of the iceberg as, in reality, many cases are kept secret and victims accept the blame for the violence [6]. Since most of this felony happens between a husband and a wife within the four walls of a home, this is usually viewed as a private matter, and they believe that what happens in the house should not be shared with others. Disclosure or help-seeking depends on the ability of a person to define an action as abusive [6].

Violence by an intimate partner is common in all ages. However, the young adult seems to be particularly at risk and this group has received more attention in recent times [7,8]. Young adulthood is a vital era when people begin to explore meaningful relationships, hence IPV is frequent in young adult societies. Young people have a good awareness of issues related to IPV; nonetheless, understanding of such matters is often low and has been influenced by their personal experiences and observations, as well as by values and norms that have been taught to them; this makes these issues appear complex [9].

The health belief model (HBM) is a well-utilized theory that examines perceptions, beliefs, attitudes, and intentions related to health behavior. The motivation to make health issues salient in a person’s life is necessary. Even though IPV intuitively does not seem like a health-related issue, it has been clearly associated with negative health outcomes. Hence, an individual needs to believe there is a perceived threat to their health [10,11,12].

Even though a growing body of research on IPV has been conducted, an understanding of this issue among the young generation has rarely been sought [13,14]. An assessment of attitudes towards IPV could have a significant positive impact on the cycle of violence. The acceptability or tolerance towards IPV has usually been linked to the perpetration of this type of violence and increases the risk of its occurrence [15]. Hence, it is valuable and essential to study the young adult’s attitudes towards IPV to provide information that could be used in violence prevention interventions. This study aimed to determine the proportion of attitudes towards rejecting IPV and its associated factors among young adults attending premarital courses in the state of Kelantan, Malaysia.

## 2. Materials and Methods

### 2.1. Study Design and Sample

This quantitative cross-sectional study was conducted from February to December 2021 and targeted premarital young adults who registered for premarital courses in four randomly selected districts in Kelantan, Malaysia. Out of 13 states in Malaysia, Kelantan was chosen as the study setting since it reported the third-highest number of reported cases of domestic violence (12%), after Selangor (14%) and Johor (12.4%) in 2017 [4]. Besides, the authors had a strong professional network in this state, facilitating the organization of the study. All Muslim couples planning to marry are required to attend a premarital course. This is a two-day course organized by the State Islamic Religious Department to provide elementary knowledge about religious aspects, stress and financial management, health topics, and problem-solving skills for marital difficulties, and which allows them to be prepared for married life and achieve the goal of marriage. The course is open to young adults as young as 18 years old; it is not limited to those who are engaged, and a single person is also allowed to attend as preparation for future marriage.

The sample size for this study was calculated as 401 using the single proportion formula. The inputs of the computation were: 95% confidence level, 5% precision, expected proportion of young adults with positive attitudes towards IPV of 32% [16], design effect of 1.96, and non-response rate of 20%. Purposive sampling was applied to select the respondents. All 421 young adults attending the premarital courses were approached. The inclusion criteria were unmarried young adults aged 18 to 30 years old. Those who did not understand the Malay language were excluded from the study. This study was approved by the Human Research Ethics Committee of our institution (USM/JEPeM/19110807).

### 2.2. Dependent Variables

Attitudes towards rejecting IPV. This refers to a good overall evaluation in rejecting IPV and the willingness to disclose it. An individual with an attitude score equal to or more than the median score was considered as having attitudes towards rejecting IPV, which was not accepting the IPV and having the willingness to disclose it.

### 2.3. Independent Variables

Independent variables in this study were those suggested by the literature as associated with attitudes towards rejecting IPV. We categorized sex as: “male” and “female”; education level as: ”primary school (up to grade 6)”, “secondary school (grade 7–10)”, “diploma (grade 11–12)”, ”university” (degree/master/doctorate level); occupational status as: “not working” (unemployed), “government worker”, “non-government worker”, and “self-employed” (working for oneself as a freelancer or the owner of a business); current relationship status as: “boyfriend or girlfriend”, “fiancé”, “single”; and sources of information regarding IPV (television news, documentaries, films, television dramas, radio, magazines, books, newspaper, pamphlets, posters, websites and social media).

### 2.4. Survey Instrument

A validated Malay Intimate Partner Violence Questionnaire (MY-PAIPVQ) was used [17]. The Cronbach’s alpha values ranged from 0.817 to 0.972 and Raykov’s Rho values were in the range of 0.613–0.982. The questionnaire consists of sections on demographic information and attitudes towards IPV.

A total of 23 items on attitudes towards IPV in the questionnaire assessed the domains of acceptance of IPV and willingness to disclose IPV. Examples of items written as negative statements included “violence is one of the ways to express the anger”, “violence is needed to solve problems”, and “someone who cheats on his/her partner deserves to be hurt”, while the positive statement was “physical violence, even though not leaving any marks, should not be accepted”. Respondents were also asked about their willingness to disclose IPV to formal persons (such as police officers, health staff, and lawyers) and informal persons (family members and friends) and the *Talian Kasih* hotline, which is provided by the Women, Family, and Community Development Ministry to provide help pertaining to women, family, and community issues.

The items were scored using the Likert Scale ranging from “strongly disagree” = 1, “disagree” = 2, “neutral” = 3, “agree” = 4, “strongly agree” = 5 for positive statements, while the score was reversed for negative statements. The final scores were categorized into two: attitudes towards rejecting IPV (attitudes score equal to or more than the median score) and attitudes towards accepting IPV (attitudes score less than the median score) towards IPV.

### 2.5. Data Collection and Procedure

In view of a movement control order (MCO) due to the COVID-19 outbreak, premarital courses were withheld. The participants were only required to register for the course and would be contacted later once the course resumed. Hence, data for this study were collected through the virtual method of a Google Form, an online survey software. The registration lists of premarital courses participants were obtained from the religious offices. The potential respondents in the list were then contacted, a briefing on this study was given and the Google form link was sent to the respondents through the WhatsApp application once consent had been given. By clicking on the link, the respondents were directed to the survey entry page, which contained information on the objectives of the survey, terms of participation, data privacy, and the consent form on the first page. Clear instructions were stated at the beginning of the questionnaire to avoid any possible doubts about the aim of the study and about how to complete the questionnaire. Subsequently, the respondents who consented were able to fill out the online questionnaire and submit their responses. Google’s invisible reCAPTCHA V3 was used to protect against bots and other automated programs [18].

### 2.6. Statistical Analysis

Survey data were analyzed using SPPS^®^ version 25 (SPSS, Chicago, IL, USA). The descriptive analyses were conducted to describe the sample. Simple and multiple binary logistic regression analyses were conducted to determine factors associated with attitudes towards rejecting IPV. All variables with p-values of less than 0.25 in bivariate analysis were included in the multivariable analysis, based on the Wald test from logistic regression by prioritizing the important variables to be further evaluated. These variables are also conceptually/clinically significant [19,20]. The model covariate was added using a backward stepwise selection process. The stepwise approach is useful because it reduces the number of predictors, reducing the multicollinearity problem and it is one of the ways to resolve overfitting. However, the selection of variables was also guided by the theory, and it fits well with the output of the backward stepwise selection method [21]. The association between independent and dependent variables was determined by adjusted odds ratios (AdjOR), with 95% confidence intervals (95% CI) and *p*-value < 0.05, to determine the statistical significance level of these factors. Any interactions between the variables in the preliminary main-effect model were investigated. All potential two-way interactions were examined. A correlation matrix and standard error were used to verify multicollinearity. The classification table, receiver operation curve (ROC), Hosmer and Lemeshow test, and percentage of correctly classified were used to determine the model’s fitness.

## 3. Results

### 3.1. Sample Characteristics

A total of 405 premarital young adults participated in this survey with a good response rate (96.0%). The sociodemographic characteristics of the respondents involved are summarized in Table 1. The mean age of young adults in this study was 24.43 years old (SD 3.46). Most of them were female (57%), had a secondary education (51%), worked in the non-government sector (39%), and had a fiancé relationship status (47%). The most popular source of information regarding IPV was social media, which was chosen by 299 respondents (73%), followed by television news chosen by 71%, television dramas with 60%, newspapers with 50%, and films with 47%.

### 3.2. Attitudes towards IPV

Overall, half of the respondents (50.4%) have an attitude towards rejecting IPV, with 58.8% of the respondents rejecting the IPV behaviors and 58.0% of them being willing to disclose and report IPV should they experience IPV in the future. The mean score for the domain of acceptance of IPV was 3.68 (SD 1.26), while the mean score for willingness to disclose IPV was 3.59 (SD 0.94). Most of the premarital young adults did not accept IPV and had the willingness to disclose IPV to the listed persons. The most popular statement for the rejection of IPV (81.7% of respondents disagreed and strongly disagreed with the statement) was “someone who cheats to his/her partner deserves to be hurt”, followed by the statement “violence is needed to solve problems” with 81.2% of respondents. However, a high number of respondents agreed with the two reverse statements: “it is a norm for someone to excessively control his/her partner” (49.6%), and “violence is just one of the ways to express anger” (61.3%). The sequence of most trusted persons to disclose IPV in the future were family members (87.9%), counselors (80.0%), employers (75.5%), and lawyers (72.3%). We also found that most young adults were not willing to disclose IPV to police officers (78.0%) and the Talian Kasih hotline (65.9%). Detailed findings on attitudes towards IPV are shown in Table 2.

### 3.3. Determinants of Attitudes towards Rejecting IPV

Table 3 presents the univariable level of logistic regression analysis of factors associated with attitudes towards rejecting IPV among premarital young adults. From the crude analysis, the variables with a p-value of less than 0.25 were age, sex, occupational status, current relationship status, and sources of information regarding IPV (television news, documentaries, films, dramas, magazines, books, newspapers, pamphlets, websites, and social media). Multivariable analysis results after the backward stepwise selection method by likelihood-ratio test, the model ended up with four significant variables. The variables were age, sex, occupational status, and drama as sources of information regarding IPV (Table 4).

A young adult with an increase of 1 year of age has 1.12 times the odds of having an attitude towards rejecting IPV (95% CI:1.03, 1.19; *p*-value = 0.003) when adjusted for other variables. Females have 2.49 times the odds compared with men of having an attitude towards rejecting IPV (95% CI:1.54, 4.03; *p*-value < 0.001) when adjusted for other variables. Those who were self-employed were 80% less likely to have an attitude towards rejecting IPV compared with young adults who were not working (95% CI:0.09, 0.40; *p*-value < 0.001) when adjusted for other variables. Young adults who receive information regarding IPV from dramas had 3.66 times the odds of having an attitude towards rejecting IPV (95% CI:2.26, 5.91; *p*-value < 0.001) compared with those who did not receive the information from drama when adjusted for other variables.

## 4. Discussion

This study utilized an online method for data collection and had a very good response rate (96.0%). Previous studies suggested an acceptable response rate of above 60% for most research types and an 80% response rate is required when intended to generalize to all populations [22]. Most of the respondents in this study had a secondary education. A possible explanation might be due to most of the respondents being in their twenties, thus they are still studying in higher education or have recently completed their secondary school education.

Five out of six items in the domain of acceptance of IPV were negative statements and were written as statements that foster IPV. Most of the participants disagreed with these items, reflecting that they rejected IPV. However, a high number of respondents agreed with two items: the statements that (1) it is a norm for someone to excessively control his/her partner and (2) violence is just one of the ways to express anger. Similar findings were reported as close to half of participants in a previous study agreed that sometimes violence is the only way to express feelings; moreover, women were also widely recognized in the literature for being violent in their relationship with men [23,24].

Regarding willingness to disclose IPV, quite a high number of respondents were unsure and disagreed with the option of disclosure to police officers (78%) or the *Talian Kasih* hotline (65%). A similar situation was reported in Nigeria where many IPV cases were not reported to the police [25]. The community sees it as a domestic affair that needs no intervention from the police. Consequently, IPV was under-reported and under-recorded by the police. In addition, a study in Washington reported that young adults prefer a telephone helpline less for safety reasons. They claimed that talking to a stranger on the telephone is risky and adds pressure and nervousness. However, the study also found that some respondents preferred the telephone helpline because it provides the option of hearing another person’s voice and it can also provide support and guidance in a non-judgmental manner [26].

Binary logistic regression analysis revealed four factors associated with attitudes towards rejecting IPV. Young adults of an older age tended to have attitudes towards rejecting IPV. This finding is also consistent with several other studies which showed that a younger age was associated with the acceptance of IPV [27,28,29,30]. Older adults have better attitudes towards IPV as reflected by the developmental shifts in attitudes and other qualities such as sensitivity and moral awareness [31,32]. Educational differences also may explain this pattern, as younger people lack exposure to the liberalizing influence of higher education experienced by older adults.

Female young adults were found to have double the odds of attitudes towards rejecting IPV compared with male respondents. This finding is parallel with findings in several other studies, which found that females had a higher intention of reporting partner violence and they believed that the police would take some action if they made a report [33,34]. This difference in sex might be related to female fear of harm to either themselves or their children, in the form of being injured or killed [35,36]. In contrast, another study found women who justified IPV were higher in number compared to men [37]. Young women accept IPV due to the predominance of a patriarchal system and cultural and social norms that value men as superior and more powerful than women. These norms and cultures subordinate women in many spheres of life, from economic independence to decision-making power [38,39].

Young adults who were self-employed were less likely to have attitudes towards rejecting IPV. This finding is in line with a study that found that self-employed persons were mostly engaged in multiple informal economic activities and had strong cultural beliefs that marriage is a life commitment, which increased their acceptance of IPV and made it hard for them to leave violent relationships [40]. Self-employed workers are those workers who work on their own account, or with one or a few partners, or in a cooperative. They need to dedicate more effort to numerous sources of income to maintain their income and savings. Therefore, being self-employed is known as a vulnerable type of employment in Malaysia. The self-employed are either not covered by work-related social protection (i.e., health insurance and pension funds) or entitled to certain employment benefits (i.e., paid vacations and sick leave). Being self-employed does not always mean a higher risk of violence acceptance, but it does make the individuals more vulnerable. It is due to a lack of nearby support which could influence the environment in which they operate [41,42].

Receiving information regarding IPV from dramas was associated with attitudes towards rejecting IPV. Our findings contradict research showing respondents were desensitized to violence and had been acclimated to violence, having been influenced by the depiction of violence in slasher films and dramas [43]. A study in Texas found that even though drama genres tended to show images exaggeratedly and reinforced social norms about sex, many dramas also portray IPV as an unacceptable issue that may lead to better perceptions for the viewers [44]. It is likely that this situation also occurs in Malaysia, in that the dramas were able to show that IPV is not acceptable. However, to assess the degree of exposure to violence in movies or dramas, a deeper observation including the frequency of viewing, and direct response after viewing, is required.

Although our study has a number of strengths, it also comes with a number of limitations and challenges. The main strength is that it is one of the few works to evaluate the attitudes toward rejecting IPV using a validated questionnaire. Additionally, the results obtained in this study may lead to the design of preventive actions on IPV and to the use of this assessment tool to assess attitudes towards IPV. The main limitation is the high homogeneity of the sample studied. We recommend for future research expand the subjects to include other states in Malaysia, or even to the national or international level, as well to a wider population including those who do not attend the premarital course. Besides, few statements in the questionnaire are factual so this requires revision and transforming for future studies to reflect attitudes towards violence. A mixed-method approach or qualitative method is suggested for future studies, to provide deeper insights into real-world problems and gain an understanding of underlying reasons, opinions, and motivations.

## 5. Conclusions

Premarital young adults in this study had relatively negative attitudes towards rejecting IPV. The factors associated with attitudes towards rejecting IPV were older age, being female, self-employed, and drama as the source of information. Generally, acceptance and justification of IPV were mainly associated with gender roles, norms, and socialization, particularly as regards women’s offenses and men’s role to discipline their intimate partners for their misbehaviors. This study cannot resolve the cultural problems of beliefs that accept IPV or increase the willingness to disclose IPV; rather this study made an effort to understand the premarital young adult’s perspective regarding attitudes towards rejecting IPV issue. Therefore, the identification of factors associated with attitudes towards rejecting IPV may enable and empower them to recognize abusive behavior is taking place and how to take appropriate action against it. We plan for a further intervention study for similar subjects in developing an educative module using language that young adults find relevant.

## Figures and Tables

**Table 1 ijerph-19-05718-t001:** Characteristics of the respondents (n = 405).

Variables	n (%)	Mean (SD)
Age (years)		24.43 (3.46)
Sex		
Male	173 (42.7)	
Female	232 (57.3)	
Educational level		
Primary School	7 (1.7)	
Secondary School	208 (51.4)	
Diploma	99 (24.4)	
University	91 (22.5)	
Occupational status		
Not working	89 (22.0)	
Government worker	31 (7.7)	
Non-Government	160 (39.5)	
Self-employed	125 (30.8)	
Current relationship status		
Boyfriend/girlfriend	149 (36.8)	
Fiancé	192 (47.4)	
Single	64 (15.8)	
Source of information regarding IPV		
Television news	290 (71.6)	
Documentaries	101 (24.9)	
Films	193 (47.7)	
Dramas	245 (60.5)	
Radio	102 (25.2)	
Magazines	98 (24.2)	
Book	52 (12.8)	
Newspaper	206 (50.9)	
Pamphlets	41 (10.1)	
Posters	40 (9.9)	
Websites	160 (39.5)	
Social media	299 (73.8)	

**Table 2 ijerph-19-05718-t002:** Attitudes toward IPV domain in MY-PAIPVQ (n = 405).

Items	Answer response, n (%)				
	Strongly Agree	Agree	Neutral	Disagree	Strongly Disagree
**Acceptance of IPV**					
It’s a norm for someone to excessively control his/her partner *	35 (8.6)	166 (41.0)	78 (19.3)	93 (23.0)	33 (8.1)
Violence is one of the ways to express anger *	97 (24.0)	151 (37.3)	56 (13.8)	75 (18.5)	26 (6.4)
Violence is needed to solve problems *	17 (4.2)	25 (6.2)	34 (8.4)	156 (38.5)	173 (42.7)
Someone who cheats on his/her partner deserves to be hurt *	6 (1.5)	22 (5.4)	46 (11.4)	183 (45.2)	148 (36.5)
Violence is the only way to discipline his/her partner *	12 (3.0)	48 (11.9)	68 (16.8)	158 (39.0)	119 (29.4)
Physical violence is not acceptable even not left any violent mark on him/her	79 (19.5)	116 (28.6)	88 (21.7)	80 (19.8)	42 (10.4)
**Willingness to disclose IPV**					
I will disclose this to friends	30 (7.4)	151 (37.3)	111 (27.4)	99 (24.4)	14 (3.5)
I will disclose this to family members	155 (38.3)	201 (49.6)	32 (7.9)	13 (3.2)	4 (1.0)
I will disclose this to health staff (doctors/nurses)	39 (9.6)	144 (35.6)	137 (33.8)	66 (16.3)	19 (4.7)
I will disclose this to my employer if I’m working	92 (22.7)	214 (52.8)	67 (16.5)	23 (5.7)	9 (2.2)
I will disclose this to counselors	128 (31.6)	196 (48.4)	57 (14.1)	16 (4.0)	8 (2.0)
I will disclose this to religious officers	100 (24.7)	156 (38.5)	114 (28.1)	25 (6.2)	10 (2.5)
I will disclose this to lawyers	109 (26.9)	184 (45.4)	80 (19.8)	21 (5.2)	11 (2.7)
I will disclose this to police officers	21 (5.2)	68 (16.8)	167 (41.2)	111 (27.4)	38 (9.4)
I will disclose this to the Talian Kasih hotline	34 (8.4)	104 (25.7)	185 (45.7)	62 (15.3)	20 (4.9)

* Statement * = reverse statement.

**Table 3 ijerph-19-05718-t003:** Simple logistic regression of the association between factors and attitude towards rejecting IPV among premarital young adults in Kelantan (n = 405).

Variables	Crude OR (95% CI)	*p*-Value
Age (years)	1.08 (1.02,1.15)	0.005 *
Sex		
Male	1	
Female	2.82 (1.87,4.24)	<0.001 *
Educational level		
Primary School	1	
Secondary School	1.87 (1.20,2.03)	0.992
Diploma	2.01 (1.93,2.54)	0.762
Degree/Master/PhD	1.81 (1.20,1.99)	0.846
Occupational status		
Not working	1	
Government worker	1.12 (0.48,2.63)	0.788
Non-Government	0.90 (0.53,0.97)	0.708
Self-employed	0.23 (0.12,0.41)	<0.001 *
Current relationship status		
Boyfriend/girlfriend	1	
Fiancé	1.11 (1.02,1.70)	0.628
Single	0.61 (0.33,0.81)	0.111 *
Source of information regarding IPV		
Television news	2.57 (1.63,4.03)	<0.001 *
Documentaries	1.91 (1.20,3.04)	0.006 *
Films	2.72 (1.82,4.07)	<0.001 *
Dramas	3.72 (2.43,5.68)	<0.001 *
Radio	1.27 (0.81,2.00)	0.290
Magazines	1.98 (1.24,3.17)	0.004 *
Book	1.53 (0.85,2.77)	0.155 *
Newspaper	2.07 (1.39,3.07)	<0.001 *
Pamphlets	1.81 (0.92,3.53)	0.081 *
Posters	1.37 (0.71,2.66)	0.344
Websites	2.57 (1.70,3.88)	<0.001 *
Social media	2.49 (1.57,3.95)	<0.001 *

Note: OR = odds ratio; CI = confidence interval. * Variables with a *p* < 0.25 were fit into model in multiple logistic regression.

**Table 4 ijerph-19-05718-t004:** Multiple logistic regression of the association between factors and attitudes towards rejecting IPV among premarital young adults in Kelantan (n = 405).

Variables	B	Adjusted OR (95% CI)	*p*-Value
Age (years)	0.110	1.12 (1.03, 1.19)	0.003
Sex			
Male	1		
Female	0.914	2.49 (1.54, 4.03)	<0.001
Occupational status			
Not working		1	
Government worker	0.037	1.04 (1.01, 2.80)	0.942
Non-Government	0.450	0.64 (0.34, 0.84)	0.107
Self-employed	1.606	0.20 (0.09, 0.40)	<0.001
Source of information regarding IPV			
Dramas			
No		1	
Yes	1.299	3.66 (2.26, 5.91)	<0.001

Note: OR = odds ratio; CI = confidence interval. The backward LR method was applied. There were no significant interactions between the independent variables. No multicollinearity is present in the model. Hosmer–Lemeshow test, *p*-value = 0.143. Classification table 72.6% correctly classified. Area under Receiver Operating Characteristics (ROC) curve was 0.785 (95% CI:0.740,0.829).

## Data Availability

The data presented in this study are available upon request from the corresponding author.

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
