# Peer review of "Factors Associated with Attitudes towards Rejecting Intimate Partner Violence among Young Adults in Malaysia"

_ijerph, 2022, doi:10.3390/ijerph19095718_

Round 1
Reviewer 1 Report
This paper examines young adults’ attitudes toward IPV in Malaysia. The setting is unique, and the question is important. The paper has many positive aspects and could make a solid contribution. Several important limitations dampen the paper’s publishability. These are listed below.
- The prose is repetitive at times, particularly in the introduction. Please consider ways to be more concise.
- There is no theory advanced or presented.
- Why was Kelantan chosen? The fact that it has the third highest rate of domestic violence seems arbitrary. If it was chosen because the authors’ could use their professional networks there most easily, that’s fine, but say that.
- Your description of how you chose your sample size is unclear. You state the parameters used but not the actual numbers.
- Please describe your variables better. One should be able to replicate your results if they chose. Currently, that would be very difficult.
- Please be specific about the type of logistic regression model used. I’m assuming you mean binary logistic regression?
- What measures did you take to ensure the quality of your Google Forms data? Hod did you ensure that all responses were from real people rather than BOTS?
- I’d like more information on what ‘doubts’ the respondents were supposed to clear from their minds prior to taking the survey. What were these doubts and how confident are you that asking them to clear such doubts is an effective remedy for your concerns about these doubts?
- Your approach of excluding variables with p-values higher than .25 in the multivariate models is both statistically invalid because it ignores relationships between variables are multiply determined (i.e., influenced by several variables at a time) and conceptually inconsistent. If there’s a conceptual or theoretical reason to expect a variable to matter, it should be included in all analyses, no exceptions. Removing variables only serves to artificially deflate the other p-values by increasing the size of your coefficients. Do not do this.
- Table 1 makes it appear as if you had no missing data. But this seems unlikely. How did you handle missing data?
- The ‘backward stepwise selection method’ is not a valid technique for evaluating models because it a) is atheoretical and b) amplifies sampling bias. Please be theory-driven rather than data driven.
Author Response
Dear Reviewer,
Thank you for your response regarding the review of our manuscript.
The attached Appendix is the summary of the revision that I have made in our manuscript based on the reviews on the email I received dated 20th of March 2022. Please see the attachment.
I hope that our revised manuscript will be reviewed as soon as possible and hope to hear feedback from you soon.
Thank you.
Sincerely,
(DR TENGKU ALINA BT TENGKU ISMAIL)
Department of Community Medicine,
School of Medical Science,
Health Campus,
Universiti Sains Malaysia.

Reviewer 2 Report
The article is written logically and it provides valuable knowledge regarding attitudes towards Rejecting Intimate Partner Violence. I have no doubts as to the correctness of the text. However, correctly and schematically written articles always provoke the question of their meaning. On the one hand, the reader can find out what factors determine the desired and undesired attitudes, on the other hand, the question is how these factors can be modified. The independent variables have been chosen in such a way that it is practically impossible to draw any practical conclusions. It is much more difficult to ask about the specific cultural and educational factors, conditioned attitudes towards Intimate Partner Violence, that can then be modified in social campaigns or educational processes. In conclusion, given how the research was planned and conducted, the article cannot be improved. However, I am convinced that the text can show the limited importance of this research and at the same time indicate what should be the subject of research in subsequent projects, so that the knowledge provided in the research can be used in specific preventive projects and school education.
Author Response

(The authors gave the same response as above.)

Reviewer 3 Report
This paper describes attitudes towards IPV in a young population and the influence of sociodemographic variables and information sources on these attitudes. The interest of this work lies in the assessment of these attitudes in this specific sector of the population, before marriage.
CRITICISM OF THE QUESTIONNAIRE: The use of an instrument designed specifically for the Malaysian population makes it difficult to compare attitudes with other countries. At the same time, it should be noted that there is one item that expresses attitudes with difficulty, namely "Violence is one of the ways to excpress anger". Many studies support the idea that violence is often a form of expression of anger, so this item does not express violence in itself, it is a real fact. I recommend for future studies that it be transformed into "violence is an appropriate way of expressing anger", which, on the contrary, does reflect attitudes towards violence. These aspects should be included in the limitations of this study.
REDEFINE VARIABLES: It is rare in the scientific literature to use the term "good attitudes", it is a value judgment and not a scientific term. Although I also agree that rejecting violence is good. It would be more correct to use the term "attitudes of rejection towards violence", and thus try to predict the variables that influence the acceptance versus rejection of violence. This variable (good attitudes) should be redefine.
TABLES: Table 2 should be reformatted, since there is no clear correspondence of the data with each item.
SAMPLE: How is it possible that there are single people attending a marriage preparation course? what is the difference between being engaged or engaged when attending a premarital training course? Is this difference clear to respondents when answering?
STATISTICS. The type of analysis used is appropriate. What type of logistic regression do you use, binomial or multinomial? What are the DV categories? It must be specified. Since it uses logistic regression, it would be useful to add more information on the model fit ( the Hosmer and Lemeshow test, indices such as Cox and Snell's R2 and Nayelkerke's R2 ) and the % of correctly classified cases to each category.
DISCUSSION: It is worth reflecting on how attending a marriage preparation course may bias the answers to the questionnaires. For future research, it is useful to compare young people who attend these courses with those who do not.
It is poorly justified why the self-employed have higher attitudes of acceptance of violence. Could it be a cultural aspect? That is, specific characteristics of self-employment in Malaysia. Is it supported in the international literature?
The question on "receiving information about violence in dramas" is perhaps not a good indicator of exposure to violence in dramas. To assess the degree of exposure to violence in movies or dramas, one should ask about the frequency of viewing movies about intimate partner violence, or record this aspect through observation.
Author Response

(The authors gave the same response as above.)

Round 2
Reviewer 3 Report
The paper has been improved. The shortcomings han been corrected.
Author Response
Dear reviewer,
Based on the latest comments for revision that I received, I believe that you are happy with the first revision and there is no further correction/revision needed. Please confirm.